# The DNA Recognition Motif of GapR Has an Intrinsic DNA Binding Preference towards AT-rich DNA

**DOI:** 10.3390/molecules26195776

**Published:** 2021-09-24

**Authors:** Qian Huang, Bo Duan, Zhi Qu, Shilong Fan, Bin Xia

**Affiliations:** 1Beijing Nuclear Magnetic Resonance Center, School of Life Sciences, College of Chemistry and Molecular Engineering, Peking University, Beijing 100871, China; 1401110451@pku.edu.cn (Q.H.); duanbo@pku.edu.cn (B.D.); quzhi.quer@pku.edu.cn (Z.Q.); 2The Technology Center for Protein Sciences, Tsinghua University, Beijing 100084, China

**Keywords:** GapR, nucleoid-associated protein, DNA recognition mechanism

## Abstract

The nucleoid-associated protein GapR found in *Caulobacter crescent**us* is crucial for DNA replication, transcription, and cell division. Associated with overtwisted DNA in front of replication forks and the 3′ end of highly-expressed genes, GapR can stimulate gyrase and topo IV to relax (+) supercoils, thus facilitating the movement of the replication and transcription machines. GapR forms a dimer-of-dimers structure in solution that can exist in either an open or a closed conformation. It initially binds DNA through the open conformation and then undergoes structural rearrangement to form a closed tetramer, with DNA wrapped in the central channel. Here, we show that the DNA binding domain of GapR (residues 1–72, GapR^ΔC17^) exists as a dimer in solution and adopts the same fold as the two dimer units in the full-length tetrameric protein. It binds DNA at the minor groove and reads the spatial distribution of DNA phosphate groups through a lysine/arginine network, with a preference towards AT-rich overtwisted DNA. These findings indicate that the dimer unit of GapR has an intrinsic DNA binding preference. Thus, at the initial binding step, the open tetramer of GapR with two relatively independent dimer units can be more efficiently recruited to overtwisted regions.

## 1. Introduction

GapR is a nucleoid-associated protein (NAP) that is highly conserved throughout the α-proteobacteria. In *Caulobacter crescentus*, GapR is essential for normal growth and cell division [1]. Depleting GapR leads to defects in DNA replication, chromosome segregation, and cell division, while constitutive expression of GapR is lethal [2,3].

GapR shows a preference towards AT-rich DNA and overtwisted DNA, with a cell-cycle-dependent and transcription-activity-related dynamic chromosome-binding pattern [1,3,4]. It is more enriched around the origin of replication at the swarmer cell stage and moves with the replication fork after chromosome replication initiates [2]. It is also associated with highly expressed genes and operons [3]. Both the DNA synthesis inhibitor novobiocin and RNA polymerase inhibitor rifampicin can lead to the redistribution of GapR within minutes [2]. The binding of GapR to overtwisted DNA sequences in front of the replication forks or downstream highly transcribed genes stimulates gyrase and topo IV to remove the (+) supercoils that arise during the unwinding of the DNA duplex, facilitating replication and transcription [3,5].

Monica et al. firstly determined the crystal structure of GapR in complex with DNA [3]. They found that GapR can form a dimer-of-dimers that exists as a closed ring and fully encircles DNA in the middle. This structure suggested that GapR might be only able to recognize overtwisted DNA, as the larger diameter of B-DNA cannot be accommodated. Moreover, as it is implausible for the circular genomic DNA to enter into a closed ring, they proposed that free GapR exists as a dimer, and it can track along DNA and search for sites with narrow minor and expanded major grooves, where it reorganizes with another dimer to form a closed tetramer with DNA in the center [3]. However, several later studies have found that GapR remains a tetramer in solution even when DNA is absent [6,7,8]. Michael et al. proposed another model for GapR binding to DNA, in which GapR tetramer adopts an open conformation when it initially binds to DNA and undergoes structural rearrangement to fully encircle the DNA [6]. The second binding model is supported by our findings that free GapR adopts multiple conformations in solution, which are in dynamic exchange equilibrium, and it indeed adopted an open tetramer conformation in another crystal structure in complex with DNA [7]. Michael et al. also found that, besides overtwisted DNA, GapR can also bind to B-form DNA, as the central channel size is slightly adjustable. They proposed that GapR tetramer may scan along DNA and search for overtwisted high-binding sites [6].

However, why GapR prefers to bind overtwisted DNA has not been fully elucidated, although the size of the central channel may provide an obscure explain. Besides, when GapR gets released from DNA by the moving RNA/DNA polymerase, it would be less efficient for it to randomly bind to another region of the chromosome and scan for new functional sites as a closed tetramer, as the moving of GapR may be hindered by other nucleoid-associated proteins and the complex structure of the chromosome.

Here, we show that a GapR truncation mutant GapR^ΔC17^, which represents the DNA recognition motif of full-length GapR and exists as a dimer, exhibits a DNA binding preference similar to full-length GapR. We determined the crystal structure of GapR^ΔC17^ and illuminated its DNA recognition mechanism through NMR titration-based Haddock modeling. These findings indicate that the dimer unit of GapR has an intrinsic DNA binding preference towards AT-rich overtwisted DNA, which does not need the formation of a closed tetramer. It suggests that, at the initial binding step, as an open tetramer with two relatively independent dimer units, free GapR can efficiently target to overtwisted regions.

## 2. Results

### 2.1. Dimeric GapR^ΔC17^ Has the Same Fold as the Dimer Units of Full-Length Tetramer GapR

In a previous study, we found that GapR^ΔC17^ (residues 1–72, 8.2 kDa), with most of the α3 helix residues deleted, exists as a dimer in solution, as the molecular-weight values for GapR^ΔC17^ determined by using analytical ultracentrifugation (AUC) and size-exclusion chromatography coupled to multi-angle light-scattering (SEC-MALS) are 19.4 and 19.8 kDa, respectively. Moreover, a dimer, but not tetramer band, was detected in the chemical cross-linking experiment [7].

Here, we have determined the crystal structure of GapR^ΔC17^ with a resolution of 2.08 angstrom (PDB code 6K5X), with the crystallographic data statistics summarized in Table 1. Consistent with previous studies, this structure reveals a homodimer of GapR^ΔC17^. Each monomer adopts an L-shape with two α-helices (α1, residues 14–51; α2, residues 55–64) (Figure 1A). Besides the salt bridges formed by residues R26-E47, E28-R65, and E31-K66 (Figure 1B), the dimer is mainly stabilized by intermolecular hydrophobic packing (Figure 1C). The surface on the α2 helix side is highly positively charged, while the other side is negatively charged (Figure 1D). GapR^ΔC17^ generally has the same fold as the dimer units of the full-length tetramer protein (backbone RMSD = 0.95 Å), with slight changes in the positions of two α2 helices (Figure 1E).

### 2.2. GapR^ΔC17^ Can Bind to DNA with a Preference towards AT-rich Sequences

Compared with the full-length protein, GapR^ΔC17^ contains all the residues interacting with DNA and is essentially the DNA recognition motif of GapR. Interactions between GapR^ΔC17^ and DNA were studied by using 2D ^1^H-^15^N HSQC-based NMR titration experiments, as most of the backbone NH signals of GapR^ΔC17^ could be observed and assigned (Figure 2A). Adding double-stranded 6A DNA (CGCAAAAAAGCG) molecules to GapR^ΔC17^ caused the gradual shifts of many NH signals (Figure 2B). The most significantly affected residues are L30, E31, E33, A35, E36, M38, K42, E43, V44, A46, E47, K49, V55, K59, V61, R63, R69, and R72, with combined chemical shift changes over 0.06 ppm, while the *N*-terminal region residues are less perturbed (Figure 2C).

When different DNA molecules were added, the NH signals of GapR^ΔC17^ moved almost in the same directions, and the residues referred above are always the most affected (Figure 3C,D), suggesting that the binding modes of GapR^ΔC17^ to these DNA molecules should be the same. However, DNA molecules with AT-rich sequences in the middle, such as 6A and 3AT DNA, caused much more significant chemical shift perturbations than those with GC-rich sequences in the middle, such as 3CG and 6CG DNA (Figure 3C,D), suggesting that the binding affinities of GapR^ΔC17^ towards AT-rich DNA molecules are stronger. The dissociation equilibrium constants (*K*_d_) of GapR^ΔC17^ to 6A and 6CG DNA were measured by using microscale thermophoresis (MST) experiments, which are 1.1 ± 0.1 μM and 18 ± 2 μM (Figure 3E), respectively. These results suggest that GapR^ΔC17^ has a preference towards AT-rich sequences, which is the same as the full-length tetrameric GapR protein.

### 2.3. GapR^ΔC17^ Binds DNA at the Minor Groove through Electrostatic Interactions between Its Lysine/Arginine Network and DNA Phosphate Groups

Based on the NMR titration experiments, the structure models of GapR^ΔC17^ in complex with DNA were built by using the Haddock 2.4 program [10] to elucidate the DNA binding mechanism of GapR^ΔC17^. When the crystal structure of 6A DNA (PDB code 1D98) [11] was used, the top cluster contains 39 models with an RMSD of 4.2 ± 1.2 Å (compared with the model that has the lowest Haddock score) (Table 2). These models indicate that the GapR^ΔC17^ dimer binds 6A DNA at the minor groove, with two α1 helices parallel to DNA backbones and two α2 helices parallel to the axis of DNA (Figure 3A). Residues K34, K42, and K49 from two α1 helices and residues K56, K59, R63, and K66 from two α2 helices bind to DNA phosphate groups on the two sides of the minor groove through electrostatic interactions, with no sequence-specific interactions to DNA base pairs. Although not all of these 14 residues show close contact with DNA simultaneously, they form a positively charged clamp that half wraps the DNA, which may tolerate the rotation and shift of the protein on DNA. The 39 models in the cluster do not exhibit a uniform binding pattern of these residues, and the right figure of Figure 3A shows a schematic diagram of the interactions in the representative structure with the lowest Haddock score.

For the validation of the DNA binding mode of GapR^ΔC17^, NMR titration-based competition experiments were performed with a DNA minor groove binding molecule, netropsin. When netropsin was gradually added to a sample of GapR^ΔC17^/6A complex, the NH signals of GapR^ΔC17^ shifted to the positions of free GapR^ΔC17^ (Figure 4A), indicating that netropsin strongly interferes with the interaction between GapR^ΔC17^ and 6A DNA, which supports the minor groove binding mode of GapR^ΔC17^. Moreover, the DNA binding affinities of the K34A, K42A, K49A, K56A, K59A, R63A, and K66A mutant proteins with 6A DNA were measured by using MST experiments, which are significantly lower compared with the wild-type protein (Figure 4B), indicating that all of these lysine/arginine residues are important for DNA binding. These residues are highly conserved among GapR homologs (Figure 4C).

The complex models of GapR^ΔC17^/6A DNA suggest that DNA shape and the spatial distribution of phosphate groups should be related to the binding affinity by affecting the electrostatic interaction patterns. AT-rich DNA generally have higher levels of propeller-twist [12], and overtwisting can narrow the diameter and minor groove widths of DNA [13], which might be more preferred by GapR^ΔC17^.

A 12-bp GC-rich DNA molecule 3CG (CTACGCGCGTAG) (PDB code 5MVK) [14] was also docked with GapR^ΔC17^, and the results were compared with those of 6A DNA. GapR^ΔC17^ also binds 3CG DNA at the minor groove, which is similar to 6A DNA (Figure 3B). However, the 3CG DNA is less twisted with much wider minor groove than 6A DNA (Figure 3C), which might be less matched in shape with GapR^ΔC17^, as the GapR^ΔC17^/3CG DNA models in the top cluster revealed fewer close contacts between DNA phosphate groups and the interface lysine/arginine residues, and their intermolecular electrostatic energies are generally higher than those of GapR^ΔC17^/6A DNA models (Figure 3D). These findings should explain why AT-rich overtwisted DNA sequences with narrow minor grooves are more preferred by GapR^ΔC17^.

The DNA binding mode of GapR^ΔC17^ is similar to those of the two dimer units in the open tetramer/DNA complex, which also mainly bind DNA at the minor groove. Thus, the open tetramer/DNA complex should represent a local energy minimum state in which two dimer units both adopt a high-affinity DNA binding mode, which was stabilized during crystallization.

## 3. Discussion

Binding of GapR to the overtwisted regions in front of the replication fork and RNA polymerase stimulates gyrase and topo IV to eliminate (+) supercoils, which is essential for normal replication and transcription [3]. Throughout the cell cycle, the GapR level remains relatively constant. The moving of the replication folk continually recruits GapR in the front and leaves zones of GapR depletion behind [4]. Moreover, when a transcription process ends and another transcription event occurs, GapR should reach the new locus in time. Thus, GapR shows a dynamic distribution pattern and can redistribute rapidly after antibiotic treatments targeting DNA replication and transcription [2]. How GapR quickly binds to new sites is not well understood.

It was found that the size of the central tunnel of GapR in the closed form can have minor variations, which can accommodate both B-form DNA and overtwisted DNA [6]. Moreover, in solution, the relative position of the bound DNA inside the central channel of the closed tetramer changes dynamically [7]. Thus, it was proposed that GapR slides along DNA to scan for high-affinity sites and changes its position when the structure of DNA is affected by replication or transcription. However, as the chromosome may have complicated structures organized by other NAPs [15], this diffusion might be hindered and less efficient for the global redistribution of GapR.

Here, we show that GapR^ΔC17^ prefers to bind to AT-rich DNA with narrow minor grooves, implying an alternative way for the fast redistribution of GapR. The full-length GapR has two dimer units, each representing a DNA recognition motif and having an intrinsic DNA binding preference, conferring GapR with the ability to selectively bind to AT-rich overtwisted regions in the open form, even before it fully encircles the DNA. Thus, when released into the cytoplasm by the moving replication fork or the transcription machine, GapR can redistribute to other high-affinity sites with high efficiency.

## 4. Materials and Methods

### 4.1. Protein Expression and Purification

The coding sequence for GapR^ΔC17^ was synthesized according to the *gapr* gene (CCNA_03428, UniProt) in *Caulobacter crescentus*, which was then cloned into the NdeI and XhoI sites of a pET-21a (+) vector (Novagen, Beijing, China), followed by a C-terminal His_6_-tag. Point mutations of GapR^ΔC17^ were generated by using the site-directed mutagenesis kit (SBS Genetech, Beijing, China). The plasmids were then transfected into *Escherichia coli* Rosetta (DE3) competent cells (CWBIO, Beijing, China). Bacteria were cultured in 1 L Luria-Bertani (LB) medium at 35 °C, until the optical density at 600 nm reached 0.8. Expression of unlabeled proteins was then directly induced with 0.5 mM IPTG (isopropylthio-β-D-galactoside). For the preparation of NMR samples, bacteria were centrifuged (3260 rpm, 5 min) and transferred to 500 mL ^15^N labeled or ^15^N/^13^C/^2^H labeled M9 medium and recovered for 40 min before expression induction.

After 6 h of treatment of IPTG, bacteria were harvested by centrifugation, resuspended in 30 mL lysis buffer (50 mM sodium phosphate, 1 M NaCl, and 20 mM imidazole, at pH 8.0), and then lysed by sonication. After centrifugation, the target protein in the supernatant was purified by using an Ni-NTA column (Qiagen, Hilden, Germany) by standard methods and eluted in the elution buffer (50 mM sodium phosphate, 1 M NaCl, and 250 mM imidazole, at pH 8.0). The sample was further purified by size-exclusion chromatography on a Superdex 75 column (GE Healthcare Life Sciences, USA) equilibrated in 50 mM sodium phosphate and 150 mM NaCl, at pH 7.0. Fractions corresponding to target proteins, as confirmed by SDS/PAGE (Genscript, Beijing, China), were pooled, concentrated, and stored at −80 °C.

### 4.2. Protein Crystallization

GapR^ΔC17^ was concentrated to 10 mg/mL in a buffer containing 50 mM HEPES (pH 7.0) and 300 mM NaCl and crystallized by hanging drop vapor diffusion at room temperature, using an equal volume of protein sample and crystallization solution consisting of 0.1 M sodium cacodylate (pH 6.0) and 15% PEG4000. Crystals of the selenomethionyl derivative of the construct grew under similar conditions. Crystals were frozen in liquid nitrogen after a quick soak in a cryo-protectant comprising crystallization solution with 30% glycerol for data collection.

### 4.3. Crystallographic Structure Determination

The diffraction data of native protein were collected at beamline 3W1A at Beijing Synchrotron Radiation Facility, while the diffraction data of SAD protein were collected at beamline BL17U at Shanghai Synchrotron Radiation Facility (Shanghai, China). Both the native and SAD datasets were integrated and scaled by using HKL2000 [16]. The space group identified for GapR^ΔC17^ was I4_1_22, with one molecule in the asymmetric unit of the native crystal. The Phenix v1.19.2 program (Berkeley, CA, USA) was used to locate the heavy atoms and to calculate the initial phases, leading to an interpretable electron density map. The manual model building was carried out using the program Coot [17]. Finally, a model that was refined to an R_work_/R_free_ of 25.1%/26% was obtained. Data collection and the statistics of the final model are summarized in Table 1.

### 4.4. Resonance Assignment of Backbone NH Signals of GapR^ΔC17^

The NMR sample containing 1 M ^15^N/^13^C/^2^H labeled GapR^ΔC17^ protein was prepared in 450 μL 95% H_2_O/5% D_2_O with 50 mM sodium phosphate (pH 7.0), 150 mM NaCl, 0.1% NaN_3_, 5 mM EDTA, 75 mM arginine, 75 mM glutamate, and 0.01% DSS. Then 2D ^1^H-^15^N HSQC, 3D HNCACB, 3D HN(CO)CACB, 3D HNCA, 3D HN(CO)CA, and 3D HNCO experiments were performed on a Bruker Avance 500 MHz spectrometer (Bruker, Billerica, MA, USA) with a triple-resonance cryoprobe at 298 K. NMR spectra were processed by using NMRPipe [18] and analyzed by NMRView [19]. ^1^H chemical shifts were referenced according to internal DSS, and ^15^N and ^13^C chemical shifts were referenced indirectly based on their magnetogyric ratios.

### 4.5. NMR Titration Experiments

The NMR sample of GapR^ΔC17^ used for NMR titration experiments contained 0.1 mM ^15^N labeled GapR^ΔC17^ dimer in 90% H_2_O/5% D_2_O with 50 mM sodium phosphate (pH 7.0), and 150 mM NaCl. A series of 2D ^1^H-^15^N HSQC spectra for GapR^ΔC17^ were collected with gradual addition of DNA at concentrations of 0.02, 0.05, 0.1, 0.15, 0.2, and 0.4 mM, respectively. The experiments were carried out at 298 K on a Bruker Avance 600 MHz spectrometer (Bruker, Billerica, MA, USA).

The DNA-bound state protein sample used in netropsin competition experiments contained 0.1 mM uniformly ^15^N-labeled GapR^ΔC17^ dimer, 0.1 mM 6A DNA duplex in 50 mM sodium phosphate, and 150 mM NaCl (pH 7.0) with 90% H_2_O/5% D_2_O. Two-dimensional ^1^H-^15^N HSQC spectra with 0, 0.05, or 0.1 mM netropsin were recorded on a Bruker Avance 700 MHz spectrometer (Bruker, Billerica, MA, USA).

### 4.6. MST Assay

A total of 200 nM wild-type or mutant GapR^ΔC17^ dimer was incubated in the dark with a red fluorescent dye NT-647 in a buffer with 20 mM PBS, 150 mM NaCl, and 0.02% (*w/v*) *n*-dodecyl-β-D-maltoside (pH = 7), for 30 min, at room temperature. Samples contain 20 nM labeled dimer with 16 different ratios of DNA molecules. MST assays were performed at 60% MST power and 40% excitation power, using the Monolith NT.115 instrument (NanoTemper Technologies, München, Germany). Each measurement was repeated three times. Data analyses were performed by using the MO Affinity Analysis v.2.2.4 software (München, Germany).

### 4.7. Generation of Haddock Models of GapR^ΔC17^ and DNA

Crystal structures of GapR^ΔC17^, 6A DNA, and 3CG DNA were subjected to energy minimization, using AMBER before docking. GapR^ΔC17^ residues L30, E31, E33, A35, E36, M38, K42, E43, V44, A46, E47, K49, V55, K59, V61, R63, R69, and R72 that were most significantly perturbed in the NMR titration experiments, and DNA residues 5–11 and 17–23 were defined as positive residues, in order to prevent translational displacements of GapR^ΔC17^ along DNA. Passive residues were defined automatically. Solvated docking was performed by using the guru interface of the server with the default parameters [20]. The number of structures for rigid docking was 1000. Two hundred best models were subjected to semi-flexible refinement and final refinement, which were then clustered with an RMSD cutoff of 7.5 Å and a minimum cluster size of 4.

## Figures and Tables

**Figure 1 molecules-26-05776-f001:**
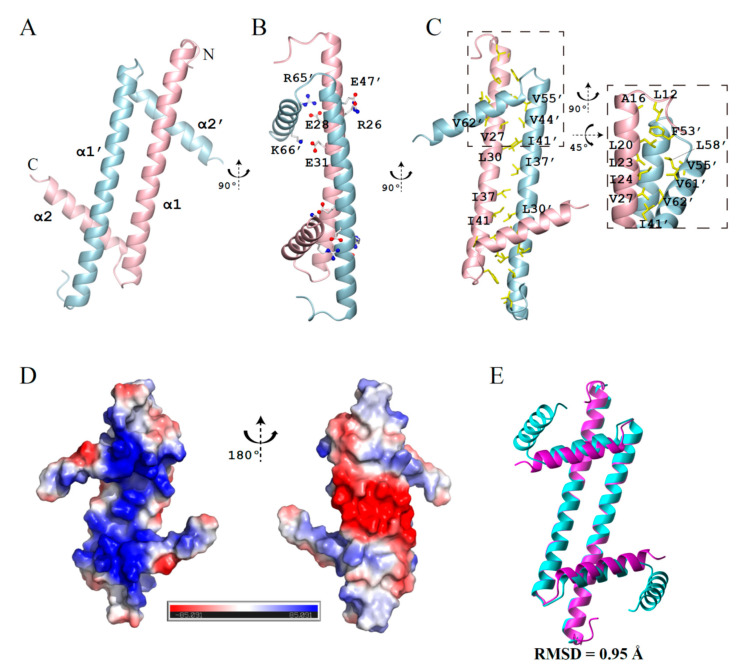
Crystal structure of GapR^ΔC17^. (**A**) Ribbon representation of GapR^ΔC17^. (**B**) Residues forming intermolecular salt bridges. (**C**) Hydrophobic residues at the dimerization interface. (**D**) The electrostatic potential surface of GapR^ΔC17^ computed by using APBS [9]. (**E**) Superposition of the GapR^ΔC17^ dimer with a dimer unit from the full-length tetrameric GapR (PDB code 6CG8).

**Figure 2 molecules-26-05776-f002:**
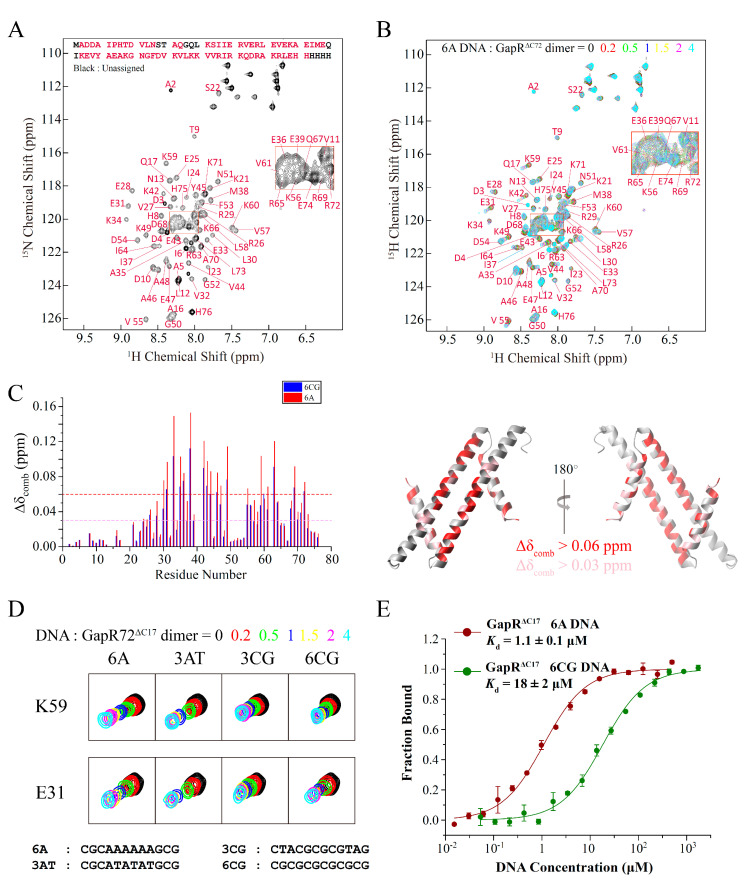
Interactions of GapR^ΔC17^ with DNA studied by NMR titration and MST experiments. (**A**) Assignment of backbone NH signals of GapR^ΔC17^. (**B**) 2D ^1^H-^15^N HSQC spectra of GapR^ΔC17^ with different ratios of 6A DNA. (**C**) Chemical shift perturbations of 4-fold 6A or 6CG DNA on the NH signals of GapR^ΔC17^. Residues with combined chemical shift changes (Δδ_comb_ = [Δδ_HN_^2^ + (Δδ_N_/6.5)^2^]^1/2^) larger than 0.06/0.03 ppm when 4-fold 6A DNA was added are colored red/pink on the structure. (**D**) Comparison of the influences of different DNA molecules on the NH signals of GapR^ΔC17^. (**E**) Binding affinities of GapR^ΔC17^ with 6A and 6CG DNA, measured by microscale thermophoresis experiments.

**Figure 3 molecules-26-05776-f003:**
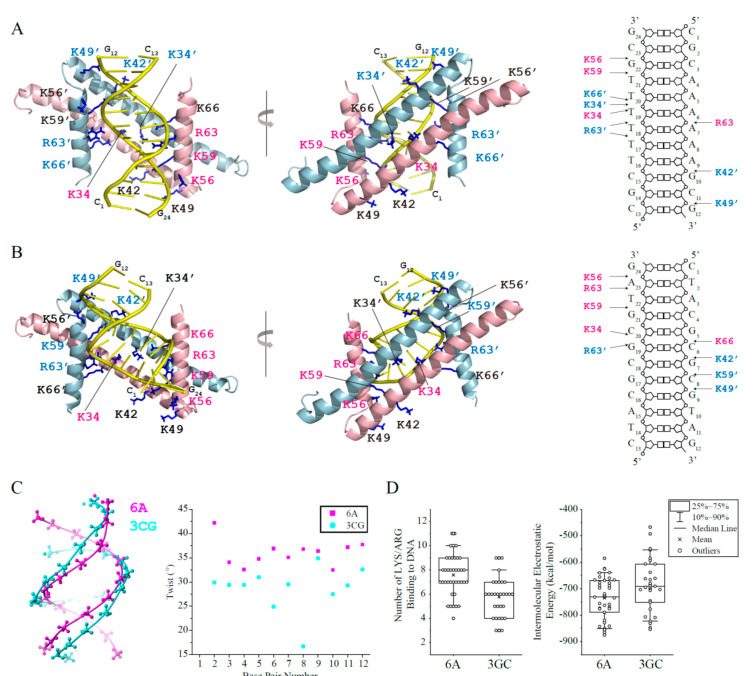
Haddock models of GapR^ΔC17^ in complex with 6A or 3CG DNA. (**A**) GapR^ΔC17^/6A complex model with the lowest Haddock score in the top cluster. The sidechains of lysine/arginine residues interacting with DNA are shown as sticks, and those close to DNA phosphate groups are colored pink (monomer 1) or light blue (monomer 2). (**B**) GapR^ΔC17^/3GC complex model with the lowest Haddock score in the top cluster. (**C**) Comparison of the structures of 6A and 3CG DNA. (**D**) For each complex model in the top Haddock clusters of GapR^ΔC17^/6A and GapR^ΔC17^/3GC, the number of arginine and lysine residues close to DNA phosphate groups (d_NE/NH1/NH2-OP1/2_ < 3.3 Å, d_NZ-OP1/2_ < 3.3 Å,) were calculated. The electrostatic energy of each complex model in the top clusters was presented below.

**Figure 4 molecules-26-05776-f004:**
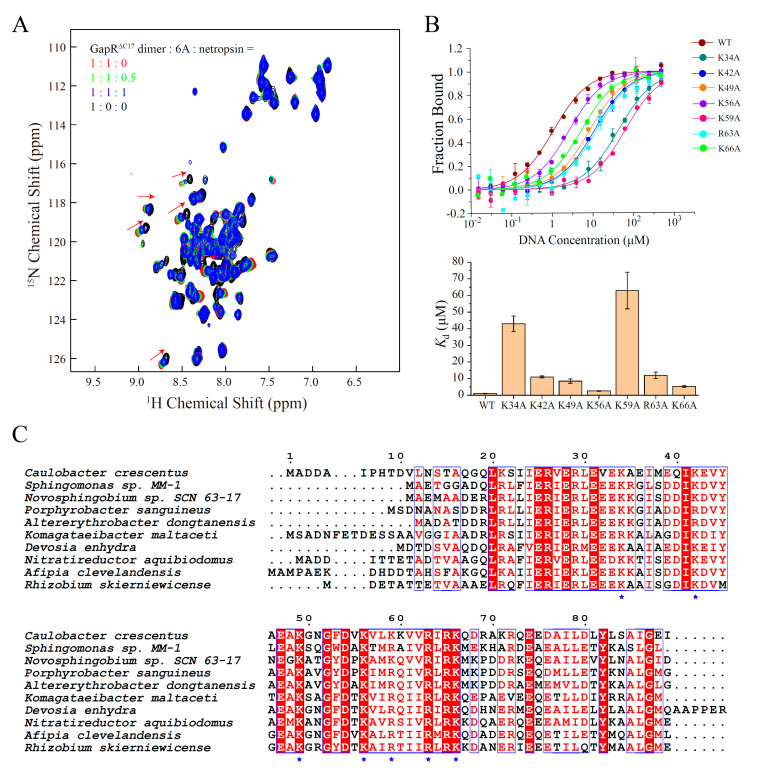
Validations of the structure model of GapR^ΔC17^/6A DNA. (**A**) 2D ^1^H-^15^N HSQC of GapR^ΔC17^ with different ratios of 6A DNA and a DNA minor groove binding molecule netropsin. (**B**) DNA binding affinities of WT and mutant GapR^ΔC17^ proteins with 6A DNA, measured by microscale thermophoresis experiments. The error bars were calculated from three independent measures. (**C**) Sequence alignment of GapR homologs from different species. The lysine/arginine residues involved in DNA binding are marked with blue stars.

**Table 1 molecules-26-05776-t001:** Data collection and refinement statistics of the crystal structure of GapR^ΔC17^.

	GapR^ΔC17^ (SeMet)	GapR^ΔC17^ (Native)
Data collection		
Space Group	I4,122	I4,122
Cell dimensions		
a, b, c (Å)	44.8, 44.8, 227.2	47.4, 47.4, 225.5
α, β, γ (º)	90, 90, 90	90, 90, 90
Resolution (Å)	56.8~3.0	47.72~2.08
Rmerge (%)	11.6 (27.1)	10.2 (53.3)
I/σI	29.28 (3.2)	48.8 (1.93)
Completeness (%)	99 (94.6)	98.06 (95.0)
Redundancy	22.4 (11.9)	22.2 (12.7)
**Refinement**		
Resolution (Å)		47.72~2.08
No. reflections		8095
Rwork/Rfree (%)		25.1/26
No. atoms		
Protein		532
B-factors		
Protein		37.0
R.m.s. deviations		
Bond lengths (Å)		0.006
Bond angles (º)		0.742
Ramachandran plot statistics		
Most favored (%)		100
Additional allowed (%)		0.0
Generously allowed (%)		0.0
Disallowed (%)		0.0

One crystal was used for each structure. Values in parentheses are for the highest resolution shell. *R**merge* = ΣhΣi|*I**h,i* −* I**h*|/ΣhΣi*I**h,i*, where *I**h* is the mean intensity of the *i* observations of symmetry-related reflections of *h*. *R* = Σ|*F**obs* −* F**calc*|/Σ*F**obs*, where *F**calc* is the calculated protein structure factor from the atomic model (Rfree was calculated with 5% of the reflections selected randomly).

**Table 2 molecules-26-05776-t002:** Statistics of the top Haddock cluster of GapR^ΔC17^ with 6A or 3GC DNA.

	GapR^ΔC17^/6A	GapR^ΔC17^/3GC
Haddock score (kcal/mol)	−344 ± 27	−335 ± 42
Cluster size	39	31
RMSD (Å)	4.2 ± 1.2	3.5 ± 2.1
E_vdw_ (kcal/mol)	−43 ± 7	−47 ± 9
E_elec_ (kcal/mol)	−735 ± 76	−686 ± 98
E_air_ (kcal/mol)	77 ± 43	109 ± 47
BSA (Å^2^)	1695 ± 138	1641 ± 186

E_vdv_, intermolecular van der Waals energy; E_elec_, intermolecular electrostatic energy; E_air_, ambiguous interaction restraints energy; BSA, buried surface area.

## Data Availability

The structure of GapR^ΔC17^ was deposited at The RCSB Protein.

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
