# Peer review of "The DNA Recognition Motif of GapR Has an Intrinsic DNA Binding Preference towards AT-rich DNA"

_molecules, 2021, doi:10.3390/molecules26195776_

Round 1

Reviewer 1 Report

A well-written manuscript describing the structural characterisation of a truncated analogue of GapR. A crystal structure of the GapR mutant was determined using X-Ray crystallography and aligned to the full-length tetramer showing similar structures. NMR was used to monitor binding to DNA, and combined with computational modelling to propose a binding model. I think this study is an excellent example of combining X-ray crystallography with NMR spectroscopy to probe the function of a protein. 

Minor comments:

  1. In section 2.2 authors reference Figure 3, but I believe they mean Figure 2. Could you authors please check.
  2. The text in several figures is very small and hard to read. Consider revising figures.

Reviewer 2 Report

This paper describes that the DNA binding domain of GapR forms a dimer and preferentially binds to minor groove of the AT-rich DNA.  The crystal structure and the binding experiments in the solution state rigidly support the binding model.  The finding that the dimer structure is the essence of preference to AT-rich sequences is new and this paper should be published.   

In 4.7 Generation of Haddock models of GapRΔC17 and DNA 

The active residues for DNA are set as 5-11 and 17-23 without explanation. When they are not set, are the same models obtained (except for simple translational displacements along DNA backbone)? 
